# Animal Models and *Helicobacter pylori* Infection

**DOI:** 10.3390/jcm11113141

**Published:** 2022-05-31

**Authors:** Shamshul Ansari, Yoshio Yamaoka

**Affiliations:** 1Department of Environmental and Preventive Medicine, Oita University Faculty of Medicine, Yufu 879-5593, Oita, Japan; ansarishamshul483@gmail.com; 2Department of Medicine, Gastroenterology and Hepatology Section, Baylor College of Medicine, Houston, TX 77030, USA; 3Institute of Tropical Disease, Universitas Airlangga, Surabaya 60115, Indonesia

**Keywords:** *Helicobacter pylori* infection, animal model, Mongolian gerbils, gastric pathogenicity

## Abstract

*Helicobacter pylori* colonize the gastric mucosa of at least half of the world’s population. Persistent infection is associated with the development of gastritis, peptic ulcer disease, and an increased risk of gastric cancer and gastric-mucosa-associated lymphoid tissue (MALT) lymphoma. In vivo studies using several animal models have provided crucial evidence for understanding the pathophysiology of *H. pylori*-associated complications. Numerous animal models, such as Mongolian gerbils, transgenic mouse models, guinea pigs, and other animals, including non-human primates, are being widely used due to their persistent association in causing gastric complications. However, finding suitable animal models for in vivo experimentation to understand the pathophysiology of gastric cancer and MALT lymphoma is a complicated task. In this review, we summarized the most appropriate and latest information in the scientific literature to understand the role and importance of *H. pylori* infection animal models.

## 1. Introduction

*Helicobacter pylori* infections are most often observed during childhood. Bacteria mainly colonize the gastric mucosa, and if they remain untreated, the infection persists throughout life. Although geographical variations in the current infection rate range from 80 to 90% to as low as 2.5%, at least 50% of the world’s population is infected with this bacterium [1,2]. Epidemiological data suggest that infection transmission occurs via oral, fecal–oral, or iatrogenic routes. Furthermore, water can be a potential source of infection transmission [3]. *H. pylori* infection is associated with the development of gastric complications, such as gastritis and gastric and duodenal ulcers, and infection increases the risk of developing gastric cancer. According to a 2018 report, approximately 13% of all global cancer cases are caused by carcinogenic infections. *H. pylori*-associated gastric cancer accounts for the largest proportion of cancer cases owing to carcinogenic infections [4,5]. It has also been suggested that over 77% of new gastric cancer cases and over 89% of new non-cardiac gastric cancer cases are associated with *H. pylori* infection [4].

In Correa’s cascade of gastric cancer development, a multistep intestinal-type process including the sequential development of chronic gastritis (CG), atrophic gastritis (AG), intestinal metaplasia (IM), dysplasia (DP), and finally gastric cancer [6,7], individuals with AG exhibit the highest risk (3.5-fold) for developing gastric cancer [8]. However, eradicating the *H. pylori* infection and the surgical resection of precancerous lesions helps prevent cancer development [9]. Although rapid progress in research on the pathogenesis of *H. pylori* infection has become possible owing to the use of animal models in recent years, it is still unknown which stage of *H. pylori* infection is involved in carcinogenesis, whether it is at an early stage, during later stages, or throughout the infection [10,11,12,13,14,15]. Furthermore, despite great efforts, it is difficult to design and establish a suitable animal model because of the complications involved in developing chronic gastric colonization in laboratory animals with *H. pylori*.

## 2. Animal Models

*H. pylori* is well adapted for colonizing the human stomach, but the natural history of infection in animals is unknown, and it does not easily infect the gastric mucosa of animals. This is because of the complex interaction of *H. pylori* with the human gastric epithelium, which takes decades to develop into gastric cancer. It is difficult to determine the pathogenesis of *H. pylori* infection and the immune response generated by this pathogen. Therefore, considerable efforts are being made to identify suitable animal models to understand the natural history of *H. pylori* infection and its immune response. It has been suggested that animals, including dogs, cats, pigs, monkeys, mice, Mongolian gerbils, and guinea pigs, could be potential habitats for *H. pylori* [16,17]. *H. pylori* does not easily infect the gastric mucosa of other animals, and in the search for more suitable animal models, experimental infection studies have been widely conducted in Mongolian gerbils, mice, guinea pigs, and rhesus monkeys.

In the human stomach, *H. pylori* mostly colonize the antrum, a pyloric part of the stomach found throughout the gastric mucosa from the pylorus to the cardia [18]. The gastric topological locations of most animal models are also preferentially colonized by *H. pylori* [19,20,21,22], emphasizing their widespread use for understanding the role of several bacterial virulence factors, host constituents, and environmental factors involved in *H. pylori*-mediated gastric pathogenicity.

Mongolian gerbils are small rodents that develop similar infection symptoms, such as appetite and weight loss, and recapitulate many features of *H. pylori*-induced gastric colonization, inflammation, ulceration, and carcinogenesis, as seen in humans [23,24,25,26,27]. Several other studies demonstrated the development of *H. pylori*-induced gastric ulcers, duodenal ulcers, and IM following an experimental bacterial challenge in Mongolian gerbils [28,29,30,31,32]. Therefore, Mongolian gerbils work as a suitable animal model and are the most commonly used animal model for establishing *H. pylori* infection. They also represent an efficient and cost-effective rodent model. Colonization of the gastric mucosa by *H. pylori* produces a similar mixed inflammatory infiltrate in the lamina propria as elicited in human diseases consisting of neutrophils and mononuclear leukocytes [33,34]. Over time, severe inflammation in gerbils causes the loss of parietal and chief cells, usually accompanied by the hyperplasia of mucous neck cells, sometimes referred to as mucous metaplasia, and the base of fundic glands may show features of spasmolytic polypeptide-expressing metaplasia (SPEM), also referred to as pseudopyloric metaplasia [33].

A mouse model infected with the *H. pylori* Sydney strain (HpSS1) displayed CG and gastric atrophy [35]. However, wild-type mouse models, such as C57BL/6 [36], BALB/c [37], and C3H [38], infected with *H. pylori* commonly develop mild gastritis or slow progressing diseases and provide less information about *H. pylori* pathogenicity [39,40,41]. Infecting mice models with *H. pylori* and *H. felis* resulted in lymphocytic gastritis without progression to severe pathologies, such as peptic ulcers or gastric cancer [22,42,43]. Moreover, the architecture of the murine stomach differs from that of the human stomach and lacks the components necessary for the development of severe gastric pathologies. Furthermore, the murine stomach may contain other bacteria that may influence the outcome of *H. pylori* infection [22,42,43]. These disadvantages limit the use of wild-type mouse models for experimental *H. pylori* infections. Therefore, several knockout or transgenic mouse models, such as insulin-gastrin (INS-GAS) [44], interferon gamma (IFN-γ), tumor necrosis factor alpha (TNF-α) knockout [45], interleukin (IL)-1 beta (IL-1β) transgenic [46], IL-10 knockout [47], Fas antigen transgenic [48], p27-deficient [49], and cytotoxin-associated gene A (CagA)-transgenic mice [50], are prone to develop gastric cancer when given a high-salt diet or chemical carcinogens of *H. pylori* infection [43,51]. Rodent animal models have been extensively used as in vivo models for studying the virulence characteristics of *H. pylori* [52,53].

*H. pylori* strains have been found to infect nonhuman primates [54,55,56,57,58], and macaques have been used as experimental models for *H. pylori* infection. However, it is unclear whether macaques carry *H. pylori* as a natural reservoir in wildlife or whether they are transmitted from humans to macaques after capture. Rhesus macaques are a suitable alternative for animal models that have several advantages over conventional small animal models, such as anatomical and physiological similarities with humans, while socially housed rhesus macaques are naturally infected with *H. pylori* [55,56]. Moreover, all infected macaques develop CG, and some may develop gastric atrophy [57], the histological precursor of gastric adenocarcinoma [59]. However, studies on non-human primates are time-consuming, tedious, labor-intensive, and extremely expensive, making it difficult to evaluate the degree of *H. pylori* virulence. Although *H. pylori* naturally infects the human gastric mucosa, observations indicate that some macaques reared in captivity were naturally infected with this bacterium [53,54,55].

The guinea pig model of *H. pylori* infection was first described in the late 1990s [20,60]. The guinea pig is a small laboratory animal with a stomach structure similar to that of humans, prone to developing an inflammatory response due to IL-8 secretion by gastric epithelial cells. Similar to the mouse model, the guinea pig models exhibit the ease of husbandry owing to the small animal size. In addition, the guinea pig stomach possesses several features in common with the human stomach, such as the presence of a cylindrical epithelium, maintenance of sterile conditions, the ability to produce IL-8, and the lack of a non-glandular region [61,62,63]. However, the studies are from 2001 to 2003, and there was no openness to metagenomics studies that show the opposite in humans, and I would suppose that in other models as well. Furthermore, like humans, guinea pigs require vitamin C [64].

These animals are considered an optimal model because they possess several similarities with human hormonal and immunologic responses, innate immunity and complement systems, and thymus, bone marrow, and pulmonary physiology. They also demonstrate a delayed type of hypersensitivity, major histocompatibility complex similarity, and possess numerous homologs of human group 1 cluster of differentiation (CD) 1 proteins and IFN-γ expression similarity [65,66,67,68,69,70,71]. These animals are not naturally infected with different Helicobacter spp., making them advantageous as an infection animal model to evaluate the role of several virulence factors in pathogenicity [72].

## 3. Animal Models in Evaluating *H. pylori*-Mediated Gastric Pathogenicity

*H. pylori* infection in the gastric mucosa of Mongolian gerbils exhibits a significant level of gastric inflammation after multiple inoculations with a high concentration of bacterial challenge, contributing to increased infection rates [73]. In one study, Mongolian gerbils developed moderate atrophy after 12 months of infection with *H. pylori* 26695, whereas no metaplasia was observed until 12 months, and light metaplasia was observed after 18 months [74]. Mongolian gerbils exhibit a similar stepwise progression of intestinal-type gastric cancer as developed in the human stomach through a cascade of well-defined pathological stages from the normal gastric mucosa to superficial non-atrophic gastritis to premalignant lesions (including AG, SPEM, IM, and dysplasia (DP)), and finally gastric adenocarcinoma [26,75,76,77]. Several studies have reported the development of gastric cancer in gerbil models infected with *H. pylori* TN2GF4, TN2, and 7.13 strains [27,77,78,79]. In our study conducted in 2007, the *H. pylori* TN2GF4 strain did not induce gastric cancer in gerbils 18 months post-infection. However, 9 and 18 months post-infection, 20% and 44% of gerbils had macroscopic gastric ulcers, respectively [79]. *H. pylori* infection activates the transcription factors nuclear factor-κB (NF-κB), interferon-sensitive response element (ISRE), activator protein 1 (AP-1), and cAMP response element-binding protein (CREB) in the gastric mucosa of gerbils [79]. A study of *H. pylori*-mediated pathogenicity utilized guinea pig models to find a suitable model for establishing gastric pathogenicity. The colonization levels 7 and 28 days post-infection were grade 1 and grade 2, respectively. All the animals that showed bacterial growth on the culture of gastric biopsies displayed gastritis 7 and 28 days post-infection, with an inflammatory cell response involving granulocytes and lymphocytes infiltrating the whole mucosa. Granulocyte and lymphocyte infiltration in gastric tissue 7 days post-infection was grade 1, whereas, after 28 days of the final challenge, granulocyte infiltration was still at grade 1, but lymphocyte infiltration increased to grade 2. Guinea pigs infected with *H. pylori* strains showed a significant increase in the number of epithelial cells when stained with an anti-Ki67 antibody compared with non-infected animals [80]. The results of this study provide evidence of *H. pylori*-induced epithelial cell proliferation.

### 3.1. Animal Models Disclosing the Role of Outer Membrane Protein Involvement

Mongolian gerbil models have provided crucial evidence on in vivo bacterial adaptation, which is important for gastric pathogenicity colonization and development (Table 1). A previous study also demonstrated changes in genetic material due to loss or acquisition via genetic recombination events [81]. The genetic changes observed in the bacterial genome due to infection of the gastric mucosa in animal models suggest a process of adaptation potentially related to the slight increase in genetic changes throughout the infection process. A study of persistent colonization conducted in the Mongolian gerbil model showed that the expression of blood group antigen-binding adhesin (BabA), an outer-membrane protein involved in adhering bacteria to the gastric mucosa, was initially increased after the infection. However, over time, expression reduced, and after 6 months, it was completely lost. Moreover, infection with *oipA* or *babA* mutant strains resulted in significantly reduced expression of cytokine levels, and the *alpAB* mutant strain did not infect the gerbils at all [82]. Another study provided evidence that in vivo bacterial adaptation can increase the virulence potential of *H. pylori* in a Mongolian gerbil model. In this study, a single gerbil was infected with human *H. pylori* strain B128, and 3 weeks post-challenge, the gerbil was sacrificed. A single colony (strain 7.13) isolated from the sacrificed gerbil’s stomach caused similar levels of inflammation to those caused by the B128 strain. However, gastric DP and adenocarcinoma developed in the gerbils infected by the 7.13 strain only, indicating that in vivo adaptation can increase the virulence potential of the strain [79]. In murine models compared to the wild-type strain, several mutations were identified in several genes, including *babA*, *tlpB*, and *gltS*, which are associated with colonization adaptation, in the post-inoculation strains. Other identified mutations were associated with chemoreceptors, pH regulators, and other outer membrane proteins involved in adaptation [83]. A study reported the adaptability of the strains in a mouse model of experimental infection with *H. pylori* [84]. The genetically adapted strains prevent the bacteria from eliciting damage to the gastric mucosa. The genetic modifications acquired for this adaptation significantly reduced inflammatory process levels in the gastric mucosa. *H. pylori* strains have also been found to switch their Lewis antigen phenotype during long-term colonization in rhesus macaques. The selection of bacterial phenotype switching helps bacteria adapt to their hosts [85]. Using macaques as an animal model, Hansen et al. suggested that both the loss of BabA expression and the overexpression of BabB can occur due to selective pressure. Evidence of BabA loss implicates OMP diversity in persistent *H. pylori* infection [86].

The involvement of *H. pylori* outer inflammatory protein A (OipA) in gastric injury was evaluated using a gerbil model. In a study conducted by Akanuma et al., the *oipA* mutant strain of *H. pylori* TN2 failed to colonize the gerbil gastric mucosa [88]. However, the *oipA* mutant strain of the *H. pylori* 7.13 strain successfully colonized the gerbil gastric mucosa [76]. In response to these conflicting results, we conducted a preliminary study and infected gerbil models with *H. pylori* TN2GF4 and 7.13 strains and their *oipA* mutant strains. The results showed that the *oipA* mutant strain of TN2GF4 did not colonize the gerbil gastric mucosa, whereas the *oipA* mutant of the 7.13 strain caused significantly lower levels of neutrophil infiltration than its parental strain and colonized successfully. Furthermore, the parental *H. pylori* 7.13 strain exhibited significantly higher inflammation scores and gerbil-specific keratinocyte chemoattractant (KC) mRNA levels than the 7.13 *oipA* mutant. Another recent study conducted by Du et al. utilizing a mouse infection model also showed that OipA, upregulating the expression of miRNA-30b and cysteine-glutamate transporter activity, could cause gastric mucosal injury in C57BL/6 male mice [89]. Using mouse infection models, the crucial role of outer membrane proteins in the colonization of the gastric mucosa was elucidated. In the study, “on” to “off” switching of outer-membrane proteins OipA, *H. pylori* outer-membrane protein (Hop)Z, HopO, and HopP influenced *H. pylori* bacterial density and colonization ability in mice. The strains with “off” switching status in two or more genes rendered a marked reduction in colonization rates compared with the strains with the “on” status. Interestingly, *oipA*-knockout mutant strains caused reduced inflammation and decreased IL-6 and KC mRNA levels [87].

### 3.2. Animal Models Disclosing the Pathogenic Role of the Cag Pathogenicity Island (cagPAI)

Different experimental animal models have provided evidence regarding adaptation-mediated genetic changes in *cag*PAI genes and their crucial involvement in gastric pathogenicity (Table 2). The Mongolian gerbil model challenged with cag-positive strains exhibited significantly more severe gastritis than those challenged with cag-negative strains [25,88,90,91,92], indicating the role of intact *cag*PAI in *H. pylori*-mediated pathogenicity. *cag*PAI encodes a bacterial T4SS, one of the widely studied *H. pylori* virulence factors, allowing the effector protein CagA to be delivered into the gastric epithelium [93,94]. CagA-positive *H. pylori* strains efficiently colonize Mongolian gerbils and maintain an intact and functional Cag T4SS, allowing us to study the role of virulence factors that induce the inflammatory response and carcinogenesis pathogenicity [95]. Several other studies using gerbil models have also emphasized *cag*PAI’s importance in inducing gastric inflammation and its clinical outcome [92,96]. In 2005, we found that *cagG* mutant strains could not infect the gerbil gastric mucosa [90]. Currently, several laboratories utilize the in vivo adapted HpSS1 strain, which possesses a non-functional *cag*PAI [97,98,99]. The original HpSS1 strain possessed a functional *cag*PAI prior to the in vivo adaptation of mice. Most *H. pylori* strains fail to elicit severe gastritis in a mouse model. However, it is worth comparing the host response generated in mouse models and humans, which might be critical to understanding *H. pylori* pathogenicity. Recent studies found that the HpSS1 parental strain possessing a functional T4SS could induce severe gastritis and an increased host response [97,98,99]. The results suggested that strains with functional T4SS can employ the same virulence mechanisms to induce pro-inflammatory cytokines and successfully translocate CagA into host cells, where it undergoes phosphorylation, which the human host utilizes. This mouse model study demonstrated that CD4 T cells and IFN-γ-dependent immune pressure select mismatch repair (MRR) variants of *cagY,* rendering the T4SS non-functional [100]. They are expressed, but many probably fail to bind the β1-integrins [101].

In a study utilizing 534 *H. pylori* strains in mouse models, 271 (51%) strains showed evidence of recombination in the *cagY* gene together with insertions/deletions or nonsynonymous changes in 13 *cag*PAI essential genes implicated in functional T4SS, most commonly in *cag5*, c*ag10*, and *cagA* genes [102]. Recombination in *cagY* is the most common mechanism for the downregulation of T4SS function during chronic infection in mouse models. T4SS loss of function was also associated with changes in other essential *cag*PAI genes [102]. This phenomenon of T4SS loss of function in the chronic infection of a wild-type mouse model limits its utilization to study *cagA*-related pathogenicity. However, transgenic mouse models provide critical information and are currently the most widely used models. Toll-like receptor (TLR) 5 (TLR5) recognizes the conserved domain, termed D1 and found in the flagellins of several pathogenic bacteria, for its activation but not in *H. pylori*. A study utilizing *Tlr5* knockout and wild-type mice demonstrated that one of the *H. pylori* T4SS components, CagL, which contains a D1-like motif, mediates binding with TLR5+ epithelial cells. It can act as a flagellin-independent potent activator of TLR5 for downstream signaling. These results indicate that TLR5 is necessary for the efficient control of the *H. pylori* infection, and CagL-activating TLR5 may modulate the immune response [103].

*H. pylori* infection in transgenic mouse models that cause CagA overexpression result in hyperproliferation of gastric epithelial cells and gastric adenocarcinoma, suggesting that this protein is an oncoprotein [50]. Experimental infections conducted in macaques have frequently shown a loss of functional T4SS due to frameshift mutations in *cagY* [53]. The loss of functional T4SS in *cagY*, occurring during the acute phase of experimental infection, is likely due to a mutation burst that facilitates *H. pylori* adapting in new hosts [104,105]. *H. pylori* strains isolated from natural infections of macaques show a functional T4SS capable of delivering CagA into host epithelial cells [106]. The retention of T4SS from naturally infected macaques indicates that *H. pylori* infection acquired at an early age does not provoke a strong inflammatory response that triggers a mutation burst and a non-functional *H. pylori* genomic sequence [107]. Moreover, a recent study isolated the Hp_TH2099 strain from a macaque infection and corroborated the idea that natural infection does not cause a mutation burst as the T4SS retains its functional status [58]. Whole-genome sequencing of Hp_TH2099 revealed that this strain belonged to hpAsia2, which possesses ABC-type Western CagA and currently contains unreported variations in EPIYA-C and CagA-multimerization (CM) sequences. The variations found in the CM regions almost abolished partitioning-defective 1b (PAR1b) binding of CagA. Thus, *H. pylori* strains isolated from macaques show low virulence owing to attenuated CagA activity [58]. Skoog et al. found that *H. pylori* strains that naturally infect socially housed monkeys retained functional T4SS capable of translocating CagA. The strains also induced IL-8 expression and were highly related to human *cag*PAI, suggesting that *H. pylori* can use similar virulence mechanisms in monkeys and humans to exhibit gastric pathogenicity, supporting their potential relevance as a model for studying *H. pylori*-related pathogenicity [106].

A Mongolian gerbil model study further evaluated the topographic location of inflammation in the stomach developed by wild-type *H. pylori* and isogenic *cagA* mutants. The resulting inflammation was primarily restricted to the gastric antrum region when infected with *cagA*-mutant strains, where inflammation was not significantly involved in the acid-secreting corpus region [96]. These results suggest that functional *cag*PAI is necessary to induce corpus-predominant gastritis, which is considered the precursor in intestinal-type gastric cancer progression. Furthermore, mouse gastric epithelial cell lines have been utilized for in vitro studies to evaluate the role of intracellular CagA phosphorylation in gastric pathogenicity, as it may provide more valuable information than human AGS cell lines. The GSM06 mouse gastric epithelial cell line was developed from transgenic mice that possessed the temperature-sensitive SV40 large T-antigen gene for an in vitro investigation into the physiological and pharmacological responses that *H. pylori* infection elicits [108]. A study utilizing the GSM06 and human AGS cell lines compared the early events of *H. pylori* infection. The results showed that similar to the events produced in human AGS cells, CagA was translocated in GSM06 cells, where it underwent phosphorylation [109].

According to the hit-and-run model of infectious carcinogenesis, an infectious agent triggers carcinogenesis during the initial stages of infection, and the infectious agent does not need to be present continuously for cancer to develop [110,111,112]. However, relatively less effort has been made to test the hit-and-run carcinogenesis model in *H. pylori* infections due to CagA or Cag T4SS at specific time points. A recent study utilized the tetracycline repressor (tetR)/tetracycline operator (tetO) system to derepress the *H. pylori cagUT* operon in a Mongolian gerbil model. It showed that derepressing Cag T4SS activity during the initial stages of *H. pylori* infection could initiate a cascade of cellular alterations, leading to gastric inflammation at later time points, along with gastric cancer development in a small proportion of animals when Cag T4SS was no longer active [113]. However, another study used Mist1-KRAS mice to examine the importance of *H. pylori* involvement in the later stages of disease progression. The results showed that sustained *H. pylori* infection, together with active Kirsten rat sarcoma virus (K-Ras) expression led to a gastric lesion with severe inflammation, altered metaplasia marker expression, and increased cell proliferation and DP compared to the lesion seen in *H. pylori* non-infected active K-Ras expressing mice [114]. Overall, the results suggest that the progression of carcinogenic lesions with metaplasia, DP, and cell proliferation depends on the sustained presence of *H. pylori* infection during the later stages of disease progression. Furthermore, a recent study using a mouse model was able to find that the persistent association of *H. pylori* bacteria in the proximity of the epithelial lining is necessary to induce the urokinase-type plasminogen activator receptor (uPAR), which is important for gastric pathogenicity development [115]. The study found that *H. pylori* infection induced uPAR expression in foveolar epithelial cells of the mouse gastric mucosa during the early course of infection.

**Table 2 jcm-11-03141-t002:** The role of Cag pathogenicity islands (PAI) in gastric pathogenicity evidenced by animal models.

Animal Models	Evidence Found	References
Mongolian gerbils	Intact *cag*PAI strains exhibit more severe gastritis compared to *cag*-negative strains	[25,88,90,91,92]
	CagA-positive strains efficiently colonize and render carcinogenicity	[95]
	CagG is important for successful colonization	[90]
	Functional *cag*PAIs are necessary to induce the corpus-predominant gastritis	[96]
	*cagA*-mutant strains cause antrum-region restricted inflammation without involving the acid-secreting corpus region	[96]
Mouse model	Functional T4SS is important for CagA-mediated virulence potential	[100]
	Chronic infection causes the recombination in CagY leading to T4SS loss of function	[102]
	CagA overexpression results in hyperproliferation of epithelial cells and gastric adenocarcinoma	[50]
Rhesus macaques	Experimental infection causes a frameshift mutation in *cagY* rendering T4SS non-functional	[53]
	Natural infection does not lead to the mutation burst and shows functional T4SS	[58,106]

*H. felis*, a close relative of the human gastric pathogen *H. pylori*, has been found to elicit high-grade dysplastic lesions in C57BL/6 mice, similar to *H. pylori*-induced gastric carcinogenesis in humans [116,117,118,119]. Myeloid differentiation primary response gene 88 (MyD88) regulates helicobacter-induced gastric cancer progression. A study utilizing MyD88 deficient (MyD88−/−) mice found that mice challenged with *H. felis* developed significant severe lesions that rapidly progressed to gastric cancer in situ compared to wild-type mice. It was suggested that MyD88−/− mice could be a better rodent animal model for Helicobacter pathogenicity, where gastric adenocarcinoma can develop within 6 months of Helicobacter infection [116].

In a similar study on the role of the gastric microbiome in gastric cancer development, the microbiome composition from MyD88−/− mice was compared to that of wild-type, Trif Lps2, and double knockout (DKO) mice. The differences in Helicobacter genotypes could influence the gastric microbiome, making it more susceptible to the development of Helicobacter infection-induced gastric cancer [120]. The results potentiated the role of the microbiome’s composition in the stomach, eliciting severe gastric complications, such as gastric cancer. In a recent study of a gerbil model for gastric cancer, *H. pylori* strains expressing high levels of Thioredoxin-1 (Trx1) caused worse tubular adenocarcinoma in a significantly higher percentage of gerbils compared with *H. pylori* expressing low Trx1 levels [121]. The findings indicate that the Mongolian gerbil model could be considered an appropriate animal model to study *H. pylori* gastric pathogenicity.

Hyperproliferative antral tumors associated with inflammation were developed by utilizing a “knock-in” mouse model of gastric cancer (gp130757F/F mouse) [122]. A homozygous mutation of the tyrosine phosphatase 2 (SHP2)/suppressor of cytokine signaling 3 (SOCS3)-binding site on the IL-6 family co-receptor gp130 could prevent SHP2 binding and block signal transduction via the rat sarcoma virus (RAS)/extracellular signal-related kinase (ERK)/AP-1-signaling cascade. Inhibiting this signaling pathway prevents AP-1 transcription factors activating target genes and augments signaling reciprocally by utilizing IL-6. The overall cascade leads to the rapid development of gastric carcinogenesis [122]. Furthermore, gp130757F/F mice are sensitized to the rapid development of distal stomach cancer, causing a loss of SHP2/ERK/AP-1 transcriptional regulation [123]. Our preliminary study utilized mice models lacking the SHP2 binding site on gp130 (gp130F759 knock-in mice) with orogastric *H. pylori* CPY2052 strain and *H. felis* infections. Two CPY2052-infected mice developed hyperplastic mucosa throughout the stomach 6 months post-infection, and one mouse developed hyperplastic tumors in the corpus. In that study, phospho-ERK was undetectable in uninfected gp130F759 knock-in mice, but was markedly increased in *H. pylori*-infected mice. Furthermore, phospho-signal transducer and activator of transcription 3 (STAT3) levels were noticeably higher in uninfected gp130F759 knock-in mice than in uninfected wild-type mice, and *H. pylori* infection further increased phosphorylated STAT3 levels compared to *H. felis*. These data indicate that maximal gastric injury due to *H. pylori* infection leads to the combined activation of STATs and ERK→AP-1. Activation of AP-1 due to c-Fos expression has been found to play a role in inducing cyclooxygenase-2 (COX2) and nitric oxide synthase (iNOS) in gastric epithelial cell inflammation [124]. 

## 4. Animal Models to Evaluate the Role of Host Factors in Pathogenicity

In addition to providing evidence of bacterial virulence factors in pathogenicity, animal models have evidenced the role of host constituents in *H. pylori*-associated gastric cancer pathogenicity (Table 3). The crucial role of IL-1β in gastric cancer pathogenicity was first described in a Mongolian gerbil model. IL-1β is a T helper (Th) type 1 (Th1) cytokine that is increased within the gastric mucosa of *H. pylori*-infected individuals [125]. In *H. pylori*-infected individuals, increased IL-1β expression due to polymorphisms in IL-1β significantly increases the risk of hypochlorhydria, gastric atrophy, and gastric adenocarcinoma [126,127,128]. In one study, *H. pylori*-infected gerbils exhibited elevated IL-1β levels, accompanied by decreased gastric acidity, compared to uninfected gerbils. Moreover, treating *H. pylori*-infected gerbils with an IL-1β antagonist abolished the loss of acid secretion, indicating IL-1β’s role in achlorhydria development in the stomach of *H. pylori*-infected gerbils [129]. In addition to chemokines and cytokines, experiments using Mongolian gerbils have provided evidence of altered expression of other inflammatory mediators, such as iNOS and COX2, following *H. pylori* infection [130,131]. Evidence of the role of NF-κB activation in *H. pylori*-induced inflammation was assessed using a gerbil model [132]. Furthermore, a recent study using the Mongolian gerbil model found that CD44 genes are crucial in the proliferation of epithelial cells and gastric cancer development in response to *H. pylori* infection [133]. CD44 is a transmembrane receptor that is crucial in epithelial cell proliferation. Changes in the gerbil gastric proteome in response to *H. pylori* infection were recently evaluated using the Mongolian gerbil model and applying novel proteomic approaches and pathway analyses. Quantitative proteomic analysis of biological samples recovered from the gerbil model showed that the quantity of several proteins was significantly altered by *H. pylori* infection [134]. Other studies using a gerbil infection model also showed that an increase in serum gastrin levels is directly related to gastric epithelial cell proliferation [95,135].

A study utilizing a mouse model found that gastric dendritic cells (DCs) in gastritis protect the gastric mucosa from *H. pylori*-induced inflammation. However, it allows persistent *H. pylori* infection. The immune-modulatory function of gastric classical DCs (cDCs), possibly via the programmed cell death protein 1 (PD-1)/programmed death-ligand 1 (PD-L1) pathway, protects the gastric mucosa against lymphocytic inflammation and precancerous mucosal changes. The results showed that gastric cDCs are key cells that fine-tune the inflammatory processes elicited by *H. pylori* infection and bacterial colonization in the *H. pylori*-infected gastric mucosa [136]. Transgenic knockout mice deficient in regulatory cytokines or regulatory T cell (Treg) activity are required to induce severe gastritis. However, this causes a complicated interpretation of the results [137,138,139,140]. miRNAs are critical in immune system regulation and carcinogenesis [141,142]. In d3Tx mice stomachs, miR-21a overexpression was found, and its expression level was correlated with inflammation and lymphoid infiltrate histological scores [143]. In a recent study utilizing d3Tx mice, miR021a, miR-21b, miR-142a, miR-150, and miR-155 expression increased gradually with inflammation and lymphomagenesis (mucosa-associated lymphoid tissue (MALT) development) [144]. These miRNAs can act synergistically on common or redundant targets and signaling pathways to promote cell survival and lymphocyte proliferation [144]. In a recent study utilizing the six-week-old female C57BL/6J mice, Ruan et. al. demonstrated that CD4+ CD8αα+ double-positive intraepithelial T lymphocytes (DP T cells) play an immunosuppressive role in *H. felis*-induced gastritis, mediating chronic inflammation and possibly affecting disease prognosis [145]. Furthermore, a recent study has implicated the role of APRIL (a proliferation inducing ligand) in the development of gastric MALT lymphoma [146]. Utilizing a C57BL/6J mice model, APRIL was found to promote B-cell infiltration in *H. pylori* and *H. felis* infected mice causing the recruitment of helper T-lymphocytes in *H. felis*-infected mice. In another study of mouse model of gastric cancer, Kim et al., depicted the role of programmed cell death 1 (PD1) and its ligand (PDL1) in *H. felis*-induced tumorigenesis [147]. In the study, the overexpression of PDL1 in gastric epithelial cells was found to promote the inflammation induced gastric tumorigenesis by suppressing tumor-infiltrating CD8^+^ T-cells.

**Table 3 jcm-11-03141-t003:** Animal models showing the role of host factors in *H. pylori*-associated gastric pathogenicity.

Animal Models	Evidence Found	References
Mongolian gerbils	Role of IL-1β in *H. pylori*-associated gastric pathogenicity	[125]
	*H. pylori* infection leads to nuclear factor-κB (NF-κB) activation in *H. pylori*-associated inflammation	[132]
	Cluster of differentiation 44 (CD44) is crucial in *H. pylori*-associated epithelial cell proliferation leading to gastric cancer development	[133]
Mouse model	Gastric dendritic cells (DCs) protect the gastric mucosa from *H. pylori*-induced inflammation	[136]
	Gastric DCs allows *H. pylori* infection to persist	[136]
	miRNAs synergistically act to promote cell survival and lymphocyte proliferation	[144]
	DP T-cells play an immunosuppressive role in *H. felis*-induced gastritis	[145]
	The APRIL (a proliferation inducing ligand) promotes B-cell infiltration and development of gastric MALT lymphoma	[146]
	PD1 and the over expression of its ligand (PDL1) promote *H. felis*-induced tumorigenesis	[147]

## 5. Animal Models to Evaluate the Role of Environmental (Dietary) Factors in Pathogenicity

It is well established that not only are bacterial virulence factors and host constituents solely responsible for the development of gastric pathogenicity in humans and animals but environmental factors in the gastric lumen, including several constituents of the diet (Table 4), are important pathogenicity risk factors [148]. Among the dietary constituents, the relationship between high salt consumption and the risk of gastric cancer pathogenicity has been widely studied in a gerbil model of *H. pylori*-induced gastric carcinogenesis. A study reported that high salt consumption and *H. pylori* infection could independently induce gastric atrophy and IM, the precancerous lesions in Mongolian gerbils [149]. In another study by Gaddy et al., Mongolian gerbils were maintained on high-salt and normal-salt diets and challenged with *H. pylori* to investigate the direct effect of high salt consumption in *H. pylori*-induced carcinogenesis in a gerbil model. As a result, *H. pylori*-infected gerbils maintained on a high-salt diet significantly developed gastric carcinoma compared to *H. pylori*-infected gerbils on a normal-salt diet [150]. Other studies have also demonstrated the direct effect of *H. pylori* infection and high salt consumption on gastric carcinogenesis in the presence of the chemical carcinogens N-methyl-N-nitrosourea (MNU) and N-methyl-N-nitro-N-nitrosoguanidine (MNNG) [151,152,153,154,155].

Another dietary factor, iron, has also been shown to increase the risk of gastric carcinogenicity [157]. In a study by Noto et al., gerbils were maintained on iron-replete and iron-depleted diets and subsequently infected with *H. pylori* [26]. Infected gerbils maintained on an iron-depleted diet developed more severe gastritis, had increased DP incidence and frequency, and increased gastric carcinoma observations compared to *H. pylori*-infected gerbils maintained on iron-replete diets [26]. The data demonstrate that a high-salt and low-iron diet can significantly increase the severity and frequency of *H. pylori*-induced gastric pathogenicity in gerbil models [26].

Among several gastric environmental factors, the role of copper was evaluated using an *H. felis*-infected C57BL/6 mouse model [156]. This study used the copper chelator tetrathiomolybdate to create copper deprivation conditions. The result showed that *H. felis* infection could significantly reduce the copper concentration in the mouse stomach without affecting circulatory copper levels. However, *H. felis* could not efficiently colonize the epithelium in copper-deprived mice, showing mild gastric damage compared to the infected mice with normal copper [156].

## 6. Animal Models to Evaluate Therapeutics against *H. pylori* Infection and Cancer Progression

In the 1990s, the standard triple-therapy (STT), which consists of protein pump inhibitors (PPI), amoxicillin, clarithromycin, or metronidazole, was developed. Owing to its substantially higher eradication rate, STT is recommended as the first-line eradication therapy for *H. pylori* [158,159]. It has been found that eradication therapy for *H. pylori* combined with endoscopic resection of early gastric cancer significantly reduces the development of metachronous gastric cancer [9]. However, in recent times, STT’s efficacy has been decreasing due to the increasing development of antimicrobial resistance, mainly to clarithromycin. Therefore, the increased demand for safe and effective non-antibiotic compounds capable of eliminating *H. pylori* has become a public concern [160,161,162]. Several studies have been conducted using animal models to evaluate the eradication efficacy of numerous compounds (Table 5). A study utilizing *H. pylori* infection mouse models evaluated the potent antimicrobial efficacy of H-002119-00-001, a β-caryophyllene. H-002119-00-001 showed potent efficacy in eradicating the bacteria in *H. pylori*-infected animals compared with the animals treated with antimicrobials [163].

Similarly, a study evaluated the role of hydrogen peroxide in eradicating *H. pylori* using a Mongolian gerbil model. The animal models were orally administered the *H. pylori* ATCC 43504 strain to successfully establish the infection. Hydrogen peroxide doses of 1 mg/mL, 2 mg/mL, and 4 mg/mL were administered after 14 days, and *H. pylori* counts were determined. There was no significant difference in the bacterial count between the control and hydrogen peroxide groups, indicating that hydrogen peroxide had eliminated the bacteria. Moreover, the bacterial counts of *H. pylori* in the triple-drug group were higher than those in the hydrogen peroxide group and lower than those in the *H. pylori*-infected control group. Overall, the results of this study indicated a higher efficacy of hydrogen peroxide in eliminating and preventing the recurrence of *H. pylori* than that of triple-drug therapy [23]. Moreover, the study found no toxicity or damage due to hydrogen peroxide in the gastric mucosa. Hydrogen peroxide can disrupt bacterial cell membranes, and the oxygen-enriched environment provided by hydrogen peroxide eradicates and prevents *H. pylori* recurrence, thus providing an attractive candidate for treating *H. pylori* infection [23].

A study utilizing transgenic FVB/N INS-GAS mice and Mongolian gerbils evaluated the role of 5-ethyl-2-hydroxybenzylamine (EtHOBA) against *H. pylori*-induced gastric cancer development [164]. EtHOBA, a potent scavenger of all dicarbonyl electrophiles that react with amines, prevents cancer development in these animal models. Similarly, a mouse model developed an in vivo activatable pH-responsive graphitic nanozyme, PtCo@Graphene (PtCo@G), to selectively treat *H. pylori*. The results showed high antibacterial activity against *H. pylori* and negligible side effects on normal tissues and other symbiotic bacteria [165]. Several other studies have also attempted to potentiate the efficacy of existing antibiotics by combining them with other compounds, as the current treatment usually requires high doses and frequent administration to succeed. A similar study proposed that an innovative mucoadhesive system (Mucolast^®^) loaded with amoxicillin and clarithromycin could improve the efficacy of treatment against *H. pylori* [166]. Treatment of *H. pylori*-infected C57BL/6 mice with Mucolast^®^ loaded with antibiotics showed superior efficacy than treatment with antibiotics only, as evidenced by the bacterial count in stomach tissues and histopathological evaluations. Similarly, another study analyzing the fecal microbiome composition in *H. pylori*-infected mice evaluated the efficacy of a gentamicin-intercalated smectite hybrid (S-GM)-based treatment [167]. The results showed that a *H. pylori* polymerase chain reaction (PCR) of the gastric mucosa was significantly lower in the STT and S-GM-based treatment group than in the non-treatment group. The results also showed that S-GM-based therapy could reduce IL-8 levels and atrophic changes in the gastric mucosa. Stool microbiome analysis revealed that mice treated with S-GM-based therapy showed microbiome diversity and abundant microorganisms at the phylum level compared to STT-treated mice. Overall, these results suggested that S-GM-based treatment may be a promising and effective therapeutic agent against *H. pylori* infection [167].

Other studies have used animal models to evaluate immunological events in the control of *H. pylori* infections. A study using a mouse model suggested that blocking the TLR4 signaling pathway could downregulate MyD88 expression; reduce NF-κB activation; increase CD4+, IL-2 receptor alpha chain (CD25+), forkhead box protein 3 (FOXP3+), and Treg numbers [168]; and consequently depress the Th1 and Th17 immune response, exacerbate *H. pylori* colonization density, and reduce the degree of inflammation in the gastric mucosa infected with *H. pylori*. As a result, the interaction between the TLR signaling pathway and Tregs might be an important factor in reducing *H. pylori* colonization and suppressing the inflammatory response. This mechanism was suggested to provide a new strategy for designing effective preventive and therapeutic treatment regimens against *H. pylori* colonization [168]. The antibacterial therapeutic potential of peptides, such as tilapia piscidin 4 (TP4), against multidrug-resistant *H. pylori* was evaluated in vivo in murine models (mice and rabbits). In this study, TP4 was found to inhibit the growth of antibiotic-sensitive and antibiotic-resistant *H. pylori* by causing membrane depolarization and the extravasation of cellular constituents. TP4 treatment suppresses the Treg subset population of pro- and anti-inflammatory cytokines. *H. pylori* maintains a high Treg subset and a low Th17/Treg ratio during gastric epithelium colonization, resulting in the expression of both pro- and anti-inflammatory cytokines [169].

## 7. Conclusions

Animal models have provided crucial information about bacterial infections and virulence factors in establishing persistent infections. Animal models have also evidenced the role of environmental and host factors in severe gastric complication development. These models are widely used to evaluate therapeutics and prevent *H. pylori* infection and gastric cancer development. However, the most suitable model to provide crucial evidence for understanding the pathophysiology of gastric cancer and gastric MALT lymphoma needs to be established.

## Figures and Tables

**Table 1 jcm-11-03141-t001:** Role of *H. pylori* and outer-membrane proteins in gastric pathogenicity in animal models.

Animal Models	Evidence Found	References
Mongolian gerbils	Shows light metaplasia after *H. pylori* infection	[74]
	Develops gastric cancer when infected with TN2GF4, TN2, and 7.13 strains	[27,77,78,79]
	Nine months and 18 months post-infection, 20% and 44% of gerbils displayed macroscopic gastric ulcers, respectively.	[79]
	Loss or acquisition of genetic material via genetic recombination	[81]
	Loss of outer membrane protein blood group antigen-binding adhesin (BabA) after six months of infection	[82]
	*alpAB* mutant did not infect the gerbil experimental models	[82]
	In vivo bacterial adaptation causes an increase in virulence potential	[79]
Mouse model	In vivo bacterial adaptation causes mutations in *babA*, *tlpB*, and *gltS*	[83]
	In vivo bacterial adaptation occurs after infection of the animal stomach	[84]
	“On” to “Off” switching of outer inflammatory protein (Oip)A, HopZ, HopO, and HopP occurs	[87]
	*oipA* knockout strains renders lower inflammation than its wild-type strain	[87]
Rhesus macaques	“On” to “Off” switching of BabA occurs	[85]
Guinea pigs	Shows significant increase in epithelial cells after *H. pylori* infection	[80]

**Table 4 jcm-11-03141-t004:** Animal models to evaluate the role of dietary factors in gastric pathogenicity.

Animal Models	Evidence Found	References
Mongolian gerbils	High salt consumption in association with *H. pylori* infection induces gastric atrophy and intestinal metaplasia (IM) development	[149]
	*H. pylori*-infected animals maintained on a high-salt diet develop gastric cancer compared to *H. pylori*-infected animals maintained on a normal-salt diet	[150]
	*H. pylori*-infected animals maintained on a low-iron diet develop more severe complications than *H. pylori*-infected animals maintained on normal-iron diet	[26]
Mouse model	Copper poverty leads to mild gastric damage and decreases the ability of *H. felis* to colonize the epithelium compared to the mice with normal copper	[156]

**Table 5 jcm-11-03141-t005:** Animal models in evaluating therapeutics.

Animal Models	Evidence Found	References
Mongolian gerbils	Hydrogen peroxide eliminates *H. pylori* and prevents *H. pylori* infection recurrence	[23]
	5-ethyl-2-hydroxybenzylamine (EtHOBA) prevents gastric cancer development	[164]
Mouse model	H-002119-00-001, a β-caryophyllene, shows a potent efficacy in bacterial eradication	[163]
	The graphitic nanozyme PtCo@Graphene (PtCo@G) exerts antibacterial activity against *H. pylori*	[165]
	A mucoadhesive system (Mucolast^®^) loaded with amoxicillin and clarithromycin improves antibacterial efficacy against *H. pylori*	[166]
	The gentamicin-intercalated smectite hybrid (S-GM) proves to be an effective therapeutic agent against *H. pylori*	[167]
	Blockage of the Toll-like receptor 4 (TLR4) signaling pathway could play a role in controlling the *H. pylori* infection	[168]
	Tilapia piscidin 4 (TP4), a peptide, inhibits the growth of antibiotic-resistant and sensitive *H. pylori*	[169]

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
