# Peer review of "Animal Models and Helicobacter pylori Infection"

_jcm, 2022, doi:10.3390/jcm11113141_

Round 1

Reviewer 1 Report

The animal model for the development of diseases associated with diseases in humans has represented a challenge for the researcher, in this review the most important aspects of the most used animal models in the development of pathologies associated with H. pylori were addressed, microorganism that inhabits the human being apparently since he walks with his two limbs. If we consider that apparently there are specific species of Helicobacter, it is even more complicated to study an animal model when we induce the infection, hoping that this model will tell the natural history of the disease.

The manuscript addresses the main problems associated with the immune response, cell signaling, recognition of the microorganism over time in the different models and terapeuticas already recognized and reported in the literature. Likewise, although there is no table of advantages and disadvantages of each of them, they are addressed. However, there are some phrases that could confuse the non-expert reader that should be modified and unified in each of animals models.

"the architecture of the murine stomach differs from that of the human stomach". However, in the Mongolian gerbil it does not mention it, in it it mentions all the observed changes and it is assumed that the same thing happens as in the human, although there is only one reference to it.

"The murine stomach is not sterile, and thus, other bacteria may colonize it and influence the outcome of H. pylori infection."

"H. pylori strains adapted to rodents often lose the functional status of the bacterial type IV secretion system (T4SS) and fail to deliver
CagA to host epithelial cells", it would be worth explaining whether this happens in all models or only in mice."  This statement is described later and the context is complete, so I suggest that they remove it here, because the idea remains unfinished, when in the Mongolian gerbil in the guinea pig it is not described

"The guinea pig stomach possesses several features in common with the human stomach, such as the presence of a cylindrical epithelium, maintenance of sterile conditions." In the wild mouse it is mentioned that it is non-sterile and in the guinea pig, the same as the human stomach, sterile conditions are maintained, this is completely erroneous information and should be modified or, failing that, mention that the studies are from 2001 to 2003 and there was no openness to metagenomic studies that show the opposite in humans and I would suppose that also in other models.

Line 124, change "These animals" to guinea pig unless it refers to all of the afore mentioned models. Also, in this paragraph I suggest getting more up-to-date reference data. Your references range from 1972 to 2008 

In section 5 entitled "Animal models to evaluate the role of environmental (dietary) factors in pathogenicity", there is a slight change in style, throughout the text they mention studies, without referring to the authors, it is not incorrect, but it is noticeable .

The conclusions are rather a reflection and may be adequate in the background. In the conclusions, the author could suggest why or not to use one or another animal model and in what type of study, why each study in the different models has given information that has led to the understanding of the development of gastric pathologies . Also, as he mentions that more studies should be done, he could mention some to be done

Author Response

"the architecture of the murine stomach differs from that of the human stomach". However, in the Mongolian gerbil it does not mention it, in it it mentions all the observed changes and it is assumed that the same thing happens as in the human, although there is only one reference to it.

Response: Thank you so much for your valuable comment. The comment has been addressed by adding reference (ref. 34).

"The murine stomach is not sterile, and thus, other bacteria may colonize it and influence the outcome of H. pylori infection."

Response: Thank you so much for your valuable comment. We have addressed the comment by making the sentence clearer.

"H. pylori strains adapted to rodents often lose the functional status of the bacterial type IV secretion system (T4SS) and fail to deliver CagA to host epithelial cells", it would be worth explaining whether this happens in all models or only in mice."  This statement is described later and the context is complete, so I suggest that they remove it here, because the idea remains unfinished, when in the Mongolian gerbil in the guinea pig it is not described

Response: Thank you so much for your comment. We have addressed the comment by removing the sentence as per your suggestion.

"The guinea pig stomach possesses several features in common with the human stomach, such as the presence of a cylindrical epithelium, maintenance of sterile conditions." In the wild mouse it is mentioned that it is non-sterile and in the guinea pig, the same as the human stomach, sterile conditions are maintained, this is completely erroneous information and should be modified or, failing that, mention that the studies are from 2001 to 2003 and there was no openness to metagenomic studies that show the opposite in humans and I would suppose that also in other models.

Response: Thank you so much for your comment. The comment has been addressed by mentioning the studies duration as per your suggestion.

Line 124, change "These animals" to guinea pig unless it refers to all of the afore mentioned models. Also, in this paragraph I suggest getting more up-to-date reference data. Your references range from 1972 to 2008 

Response: Thank you so much for your comment. “These animals” refers to all afore mentioned models. References (67 and 71) have been updated as per your suggestion.

In section 5 entitled "Animal models to evaluate the role of environmental (dietary) factors in pathogenicity", there is a slight change in style, throughout the text they mention studies, without referring to the authors, it is not incorrect, but it is noticeable .

Response: Thank you so much for your comment. The comment has been addressed as per your suggestion.

Reviewer 2 Report

Ansari and Yamaoka eloquently review animal models used to study the Helicobacter pylori infection disease pathogenesis and gastric carcinogenesis. While many other closely related topics could be included, such as metaplasia models or a more detailed discussion of the inflammatory responses to Helicobacter infection, these topics are not necessary. This review is excellent and adequately covers the extensive literature on this topic.

On lines 89-91. This sentence is confusing. Are other experimental models sterile? It would be of interest to discuss how the gastric microbiota interacts with H pylori infection.

There is an extensive body of literature using H felis in mouse models. The authors briefly discuss H felis, but it would be of interest to summarize more of these findings.

Author Response

Comment 1: On lines 89-91. This sentence is confusing. Are other experimental models sterile? It would be of interest to discuss how the gastric microbiota interacts with H pylori infection.

Response: Thank you so much for your valuable comment. The comment has been addressed. The interaction of gastric microbiota with H. pylori will be discussed exclusively in detail in our next review article.

Comment 2: There is an extensive body of literature using H felis in mouse models. The authors briefly discuss H felis, but it would be of interest to summarize more of these findings.

Response: Thank you so much for your important comment. The comment has been addressed by summarizing more of these findings (references 147-149, shown in Track-change).